# Menstrual hygiene practices among high school girls in urban areas in Northeastern Ethiopia: A neglected issue in water, sanitation, and hygiene research

Yohannes Habtegiorgis[1]☉, Tadesse Sisay[2], Helmut Kloos[3], Asmamaw Malede[2], Melaku Yalew[4], Mastewal Arefaynie[4], Yitayish Damtie[4], Bereket Kefale[4], Tesfaye Birhane Tegegne[4], Elsabeth Addisu[4], Mistir Lingerew[2], Leykun Berhanu[2], Gete Berihun[2], Tarikuwa Natnael[2], Masresha Abebe[2], Alelgne Feleke[2], Adinew Gizeyatu[2], Ayechew Ademas[2], Zinabu Fentaw[5], Tilaye Matebe Yayeh[6], Fitsum Dangura[7], Metadel Adane[2]☉*

1 Clinical Governance and Quality Improvement Unit, Aleta Wondo Primary Hospital, Aleta Wondo, Sidama Regional State, Ethiopia, 2 Department of Environmental Health, College of Medicine and Health Sciences, Wollo University, Dessie, Ethiopia, 3 Department of Biostatistics and Epidemiology, University of California, San Francisco, San Francisco, CA, United States of America, 4 Department of Reproductive and Family Health, School of Public Health, College of Medicine and Health Sciences, Wollo University, Dessie, Ethiopia, 5 Department of Epidemiology and Biostatistics, School of public Health, College of Medicine and Health Sciences, Wollo University, Dessie, Ethiopia, 6 Department of Statistics, College of Natural Sciences, Wollo University, Dessie, Ethiopia, 7 Aleta Wondo Primary Hospital, Aleta Wondo, Sidama Regional State, Ethiopia

☉ These authors contributed equally to this work.
* metadel.adane2@gmail.com

**Data Availability Statement:** All relevant data are within the paper and its Supporting Information files.

## Abstract

### Background

Poor menstrual hygiene practices influence school girls' dignity, well-being and health, school-absenteeism, academic performance, and school dropout in developing countries. Despite this, menstrual hygienic practices are not well understood and have not received proper attention by school WASH programs. Therefore, this study examined the extent of good menstrual hygiene practices and associated factors among high school girls in Dessie City, Amhara Region, northeastern Ethiopia.

### Methods

A school-based cross-sectional study was employed to examine 546 randomly selected high school students in Dessie City, northeastern Ethiopia. Pretested interviewer-administered questionnaires and a school observational checklist were used for data collection. Epi-Data Version 4.6 and the Statistical Package for the Social Sciences Version 25.0 were used for data entry and analysis, respectively. Bivariate and multivariable logistics regression analyses were employed to identify factors associated with good menstrual hygiene practices. During bivariable analysis, variables with P-values less than 0.25 were retained for multivariable analysis. In the multivariable analysis, variables with a P-value less than 0.05 were declared to be significantly associated with good menstrual hygiene practices.

**Funding:** The authors received no specific funding for this work.

**Competing interests:** The authors have declared that no competing interests exist.

**Abbreviations:** AOR, adjusted odds ratio; COR, crude odds ratio; MHPs, menstrual hygiene practices; WASH, water, sanitation and hygiene.

## Results

Of the respondents, 53.9% (95% CI [49.6, 58.2]) reported good menstrual hygiene practices. The following factors were found to be significantly associated with good menstrual hygiene practices: age range 16–19 years (AOR = 1.93, 95% CI: [1.22–3.06]); school grade level 10 (AOR = 1.90, 95% CI: [1.18–3.07]); maternal education (primary) (AOR = 3.72, 95% CI: [1.81–7.63]), maternal education (secondary) (AOR = 8.54, 95% CI: [4.18–17.44]), maternal education (college) (AOR = 6.78, 95% CI: [3.28–14.02]) respectively]; having regular menses [AOR = 1.85, 95% CI: (1.03–3.32); good knowledge regarding menstruation (AOR = 2.02, 95% CI: [1.32–3.09]); discussing menstrual hygiene with friends (AOR = 1.79, 95% CI: [1.12–2.86]), and obtaining money for pads from the family (AOR = 2.08, 95% CI: [1.15–3.78]).

## Conclusion

We found that more than half of high school girls had good menstrual hygiene practices. Factors significantly associated with good menstrual hygiene practices include high school girls age 16–18 years, girls grade level 10, maternal education being completed primary, secondary and college level, having regular menses, good knowledge regarding menstruation, discussing menstrual hygiene with friends and obtaining money for pads from the family. Therefore, educating of high school student mothers about MHP should be a priority intervention area to eliminate the problem of menstrual hygiene among daughters. Furthermore, in order to improve the MHP among high school girls, further attention is needed to improving knowledge regarding menstruation among high school girls, encouraging high school girls' families to support their daughters by buying sanitary pads and promoting discussions among friends about menstrual hygiene. Schools need to focus on making the school environment conducive to managing menstrual hygiene by increasing awareness of safe MHP and providing adequate water/sanitation facilities.

## Introduction

Adolescence is the period of transition from childhood to adulthood characterized by major physiological, mental, and social changes [1]. Beginning with menarche, menstruation (ordinarily referred to as having a menstrual period), is a major physiological change that adolescent girls must learn to manage, including healthy menstrual hygiene practices.

Adequate menstrual hygiene practices (MHPs) as defined by United Nations Children Fund (UNICEF) [2] consist of the following:

> "Women and adolescent girls use a clean menstrual management material to absorb or collect menstrual blood that can be changed in privacy as often as necessary for the duration of menstrual periods, using soap and water for washing the body as required, and having access to safe and convenient facilities to dispose of used menstrual management materials." [2].

MHPs have major health and socioeconomic implications, as indicated by their relationship with the United Nation (UN) Sustainable Development Goals (SDGs) [3]. Mostly the economic or financial status of a household determines the menstrual hygiene practice [4].

Despite the fact that menstruation is a natural process, it is linked with several misconceptions, negative attitudes, and punitive practices, all of which result in adverse health outcomes [5]. Poor MHPs influence the dignity, health, and well-being of schoolgirls in low- and middle-income countries, requiring well organized and effective water, sanitation, and hygiene (WASH) interventions [6]. Lack of good menstrual hygiene can have health consequences, including increased risk of reproductive and urinary tract infections [7–11].

Insufficient opportunities to practice healthy menstrual hygiene recently received attention as a barrier to education for girls in low- and middle-income countries [12]. Studies have noted that poor sanitation in schools and absence of access to good quality sanitary products can result in lower enrolment in schools, increased absenteeism, and dropout among girls [13–15]. The absence of sufficient water, sanitation, and hygiene make girls, as well as female instructors, miss school during menstruation [4]. An Indian study showed that nearly 50 per cent girls do not have access to a separate place for bathing or changing the menstrual absorbent [16]. A UNESCO report estimated that 1 in 10 girls in sub-Saharan Africa misses school during her menstrual cycle [17]. A study in Kenya showed that school girls had difficulties managing their menstrual periods at school, causing them to remain at home during their menstruation periods [8].

In Ethiopia, healthy menstrual practices are impeded by stigma and inadequate communication between girls and their mothers, other family members, and community members. Most girls do not converse with their mothers about menstrual hygiene due to fear of being punished and prevented from going to school [18]. Also existing sanitation conditions in many schools in Ethiopia are unsatisfactory, impacting girls' education [19].

Studies in Ethiopia show that menstruation-related problems resulted in 43.0% - 54.5% of female students being absent from school for 1 to 4 days each menstrual period [20, 21]. In one study, 57.8% of girls reported that menstruation affected their academic performance negatively since menarche, 90.0% did not feel comfortable when they came to school during menstruation, and 20.2% missed exams that coincided with their menstruation days [21].

A qualitative study regarding school absenteeism and menstruation-related problems showed that 24.7% of school girls knew one or more girls who had dropped out of school and 25.4% reported they had heard about girls who had dropped out of school [21]. Although MHP is a pressing issue, not much attention has been given to this subject and studies on menstruation and its hygienic management as well as its influence on girls' education in Ethiopia [22]. Therefore, this study was designed to address the level of menstrual hygiene practices and associated factors among high school girls in Dessie City, Amhara Region, northeastern Ethiopia. For this, two research questions were addressed.

1. What is the status of menstrual hygiene practices (MHP) among high school girls in Dessie City, Ethiopia?

2. What are the factors associated with good MHP among high school girls in Dessie City, Ethiopia?

## Methods and materials

### Study design and setting

We conducted a school-based cross-sectional study by using an interviewer- administered questionnaire and direct observations from January 27 to March 6, 2020 in five high schools of Dessie City. The interviews were conducted among female students and the observational study focused on the characteristics of the schools' WASH facilities. The interviews aimed to

capture female students' experiences of menstruation and its hygienic management and the impact of menstruation on their school activities.

Dessie City is located in Amhara Regional State, 400 km north of Addis Ababa. The city is located at an altitude between 2,250 and 2,470 meters and covers an area of 15.1 km$^2$, comprising five sub-cities. Dessie had an estimated population of 245,129 in 2017, 121,177 (49.4%) of them males and 123,952 (50.6%) females [23].

Dessie City has a total of 51 elementary and high schools, including both government and private schools; 20,062 (49.6%) female and 20,350 (50.4%) male students were registered in the 2019/20 academic year. There were 7,805 students in the 9th and 10th grades, of whom 3,759 (48.2%) were males and 4,046 (51.8%) females [24].

## Source population, inclusion and exclusion criteria

The source population was all female students in Grades 9 and 10 in Dessie City high schools between January 27 and March 6, 2020. All 9th- and 10th-grade school girls of the selected high schools in Dessie City were included. Ninth and 10th-grade school girls who were absent during data collection were excluded from the study.

## Sample size determination

The sample size was determined by using a single population proportion formula [25]

$$n_0 = \frac{Z_{\frac{\alpha}{2}}^2 (P[1-p]) * D}{w^2}$$

$Z_{\alpha/2}$ is the standard normal variable value at (1-α) % confidence level (α is 0.05 with 95%CI [confidence interval], $Z_{\alpha/2}$ = 1.96), $p$ is the expected prevalence of good MHPs at 57.0% taken from a similar study in Adama, Ethiopia [26], a margin of error ($w$) 5.0% of and a design effect ($D$) of 1.5. The calculated sample sizes became 565. The source population was less than 10,000; therefore, after considering a correction formula, the sample size was 496. Then, to compensate for non-responses, 10.0% of the sample was added to the calculated sample size giving a final sample size of 546.

## Sampling technique/procedure

A multistage sampling of the two-stage sampling design was used to select the 546 study participants. In the first stage, five schools (3 from public and 2 from private schools), namely Hotie, Kidame Gebeya, Memihr Akalewold, Catholic Kidanemihret, and Hope Enterprise were randomly selected by the lottery method. Using probability proportionate to size sampling method, the calculated sample of 546 (467 for public and 79 for private schools) was proportionally allocated to the randomly selected schools. Then each selected schools were stratified by grade level and the samples was proportionally allocated to the 9th and 10th grade levels. The proportionally allocated samples in the grade level were further proportionally allocated to each section of the respective grade level. During the second stage, study participants were selected using simple random sampling (lottery method) based on classroom attendance as the sampling frame.

## Outcome and explanatory variables

The outcome variable was the practice of menstrual hygiene (good or poor). Explanatory variables were socio-demographic and economic factors; obstetric- and gynecological factors; knowledge, and source of related information; sanitary pad-related factors; and WASH-related factors.

## Operational definitions

**Menstrual hygiene practices.**   To measure the respondent's MHPs, 11 closed-ended questions [21, 27, 28] were presented; each correct answer was assigned 1 point and each incorrect response was scored as 0.

**Good or poor menstrual hygiene practices.**   Respondents who scored equal to or above the mean value (6–11) were considered as having good MHPs whereas a score of less than the mean value (0–5) was considered as poor MHPs.

**Homemade absorbents** were defined as noncommercial sanitary materials prepared by family members or girls themselves for the purpose of practicing menstrual hygiene.

**Girl-friendly WASH facilities in school** were identified by the following characteristics [29, 30]:

- Gender-specific: well kept, safe, clean, and accessible sanitation facilities

- Availability of uninterrupted water supply for 5–7 consecutive days

- The continual availability of consumables, particularly soap, water, and culturally appropriate menstrual hygiene management materials

- Waste disposal bins inside the latrines for discarded pads or other sanitary materials

**Functional toilet**—denoted toilet facilities that were not physically broken and could be used [29].

**Partially functional toilet**—denoted toilets that could be used but had some problems with the physical infrastructure requiring repairs (e.g., deterioration of concrete, loose doors, locks, deteriorating roof) [29].

**Non-functional toilet** indicated—toilets that were so badly damaged or deteriorated that they were no longer being used (squat plate broken, door missing, etc.).

**A clean toilet**–was a facility in which toilet compartments were not smelly and were without visible feces/urine, flies or litter in or around the facility.

**A somewhat clean toilet** had some smell and/or some sign of fecal matter/urine, flies and/or litter.

**A dirty toilet** had a strong smell and/or presence of fecal matter, urine, significant fly problem and/or a large amount of litter [29].

**A well-lit toilet** was a facility in which the amount of light was essentially the same inside as outside.

**A somewhat dark toilet** was a facility with less visibility inside than outside but enough light for girls to be able to look at their uniforms and tell if there is a blood stain.

**A dark toilet** was a facility that was very dark inside, making it difficult for girls to look at their uniforms and tell if there were stained.

**Good or poor knowledge of MHPs**- refers to students' menstrual knowledge calculated from their responses to 12 knowledge-specific questions [21, 28]. Each correct response earned one point whereas any incorrect or "do not know" response received no point. Respondents who scored above the mean value (8–12) were considered to have good knowledge, whereas those who scored below the mean value (0–7) were classified as having poor knowledge.

## Data collection tool and procedure

The data were collected using a structured, interviewer-administered pre-tested questionnaire adapted from relevant literature, tools prepared by UNICEF for assessing MHPs [21, 27, 28] and from other studies conducted in Ethiopia [21, 27, 28]. It was first prepared in English and then translated into Amharic. An observational checklist was also adapted from UNICEF and

EMORY University [29] which was previously used to assess the school WASH facilities in Ethiopia [31].

Data collection was carried out by three female supervisors who are public health officers and five females who had BSc degrees in midwifery. Study participants were informed about the purpose of the study and data were collected after consent was obtained. The interviews were carried out in a private setting (girls' club rooms and in quiet corners of the school compounds) to reduce social desirability bias. Before starting the interviews, students were oriented by the trained data collectors.

Direct observations, which assessed the suitability of the WASH facilities, were carried out by data collectors in collaboration with janitors, girls' club leaders, unit leaders, and directors of the schools. Direct observations evaluated school WASH facilities in regard to their suitability for managing menses, specifically if they had girl-friendly WASH facilities, were gender-specific, well kept, safe, clean and accessible, provided uninterrupted water supply for 5–7 days, the continual availability of soap, water and culturally appropriate MHM materials and disposal waste bins inside the latrines for discarded pads and other sanitary materials [29, 30]. We used same supervisors for all the schools and the data collection were carried out at the same day by assigning one data collector for each school.

## Data quality management

To ensure the validity of the data, the instrument was carefully developed and modified by the principal investigator based on the objectives of the research. Published literature was reviewed and tools prepared as recommended by UNICEF for assessing MHPs [21, 27–29]. The questionnaire was prepared in English and translated to the local language (Amharic) by the principal investigator and translated back to English by another translator to ensure consistency.

The questionnaire was pre-tested by interviewing a sample 10% of the total sample size from Nigus Michael High School prior to the actual data collection, with the aim of increasing the validity of the survey and ensuring the students understood the questions. Then, appropriate amendments were made in the tool based on the pretest and the findings were excluded from the main study.

To enhance data quality, supervisors and data collectors received one day of intensive training by the principal investigator on the objectives of the study, the data collection instruments, data collection procedures, how to approach study subjects, and how to ensure ethical practices in the field. The collected data were checked daily for completeness, reliability, and clarity by the principal investigator. In order to make the process convenient for respondents, a specific time frame for administration of the survey questionnaire was set, offering flexible times for respondents. When a study participant declined to respond to any specific questions at any time of the interview, the response was recorded as "missing."

To verify the accuracy of data entries and minimize errors, a pre-arranged coding sheet was prepared before entering the questionnaire results into the EpiData Version 4.6. Also, after the completion of data entry, randomly selected 10% of the questionnaires were thoroughly checked for errors and inconsistencies. Using SPSS software Version 25.0, missing values and outliers were properly checked and managed. Following this, frequency distributions and cross-tabulations were examined for data cleaning before the statistical analysis was performed.

## Data analysis

Data entry was done using EpiData Version 4.6 and exported to Statistical Package for the Social Sciences (SPSS) Version 25.0 for cleaning and analysis. Descriptive and analytical

statistics were employed. Descriptive analysis was used to describe the major characteristics of the respondents. To measure the level of MHP, the first step was to record the response for each item as '1' for a correct answer and '0' for an incorrect response or do not know response; then, the sum score of practices was calculated (0–11 points). Second, the mean value was calculated and the mean score of 6 designated as the cut-off point. Respondents who scored 6–11 points were considered as having good practices and those with 0–5 points were classified as having poor practices of menstrual hygiene.

Multicollinearity of the independent variables was checked using the standard error (SE) and correlation matrix. The maximum value in SE was 0.37 and the correlation matrix showed that the Pearson correlation values for the variables were less than 0.7. These values indicate that there was no multicollinearity between the independent variables. The model fitness was checked using the Hosmer-Lemeshow test and the $P$-value was $> 0.05$, indicating good fit. The omnibus test of the model was $< 0.0001$ for all steps, indicating that our model was significantly different from the constant-only model, meaning there was a significant effect of the combined predictors on the outcome variable.

Bivariate and multivariable logistic regression analyses were carried out to identify the factors in good MHPs. Bivariate logistic regression analysis was done after dichotomizing the dependent variable by coding '1' for good and '0' for poor menstrual hygiene. The bivariate analysis identified candidate variables for the multivariable analysis. To control confounding, factors in the association variables with $p < 0.25$ were entered into the multivariable logistic regression analysis, enabling identification of factors associated with good MHPs. A $p$-value of 0.05 was used as the cut-off to declare statistical significance in the multivariable analysis. The strength of the association was measured by odds ratio with corresponding 95% confidence interval (CI).

## Ethics approval and consent to participate

Ethical clearance for the study was obtained from the Institutional Ethical Review Committee of Wollo University, College of Medicine and Health Sciences. Moreover, a letter of approval and cooperation was secured and submitted to Dessie City Education Department. Participants' involvement in the study was on a voluntary basis. Before the interviews, the objectives of the study were explained and written consent was obtained from all participants above the age of 18 years. For participants under 18 years of age, assent was obtained from their parents or guardians. Students were informed of their right to skip any question or withdraw their participation at any time. During data collection, personal identifiers such as name and phone numbers of the participants were not recorded to ensure confidentiality; instead, numbers were assigned for coding purposes. In line with ethical principles, 90 sanitary pads were distributed to girls who were using homemade absorbents or none at all, and to the study participants who complained about the unaffordability of pads. The sanitary pads distribution was performed after the interview completed.

## Results

### Socio-demographic and economic characteristics of the study participants

Of the total 546 study subjects, 536 completed the interview and responded to all questions (98.2% response rate). Most of the participants 457 (85.3%) went to government-owned schools. The participants' ages were between 13 and 19 years, with a mean age of 15.7 and SD of ± 0.9 years. Of the total number of participants, 328 (61.2%) were Grade 9 students. Regarding religion, 294 (54.8%) participants were Muslims and 226 (42.2%) were Orthodox Christians.

About three-fourth of the respondents (*n* = 389, 72.6%) lived with both their mother and father. Regarding respondents' maternal education, 139 (25.9%) had reached the secondary level. Most of the respondents' mothers (*n* = 309, 57.7%) were housewives. More than half (*n* = 296, 55.2%) of the girls did not regularly receive pocket money from their families (Table 1).

### Gynecological characteristics

The timing of the onset of menses experienced by most of the respondents ranged from 11 to 16 years. Mean menarche age was 13.7 with SD of ± 0.92 years. In terms of regularity of menses, 464 (86.6%) of the respondents had a regular cycle of menses. Pain during menstruation was reported by 299 (55.8%) of the girls (Table 2).

### Knowledge and awareness regarding menstruation

Most of the participants (*n* = 500, 93.3%) had information about menstruation before attaining menarche. Menstruation was said to be a normal physiological process and hormonal effect by 505 (94.2%) and 343 (64.0%) of the respondents, respectively. Three hundred four (56.7%) of the study subjects knew that the uterus is the source of menstrual blood and 416 (77.6%) said that menstruation is a lifelong process (Table 3).

The mean score for the knowledge-based answers was 8.24, with SD of ±1.85. The overall knowledge status of the participants showed that 366 (68.3%) with 95% CI (64.2, 72.2%) and 170 (31.7%) with 95% CI (27.8, 35.8%) had good and poor knowledge, respectively.

### Source of information and communication about menstruation

Less than half (*n* = 209, 41.8%) of the respondents got information about menarche from their mothers and 413 (77.1%) discussed menstrual hygiene with their friends. Regarding open communication about menstruation within the family, 163 (30.4%) of the girls had no communication with any family member. Menstruation was kept as a secret by 80 (49.1%), and 71 (43.6%) reported that it was considered to be a shameful issue in the family (Table 4).

### Water, sanitation, and hygiene-related factors

Three-fourths 398 (74.2%) of the study participants stated that running water had been available at the school compound for between 5 and 7 days per week during the month preceding the survey; of the remaining, 91 (17.0%), 47 (8.8%) reported that water had been available for at-least 2 to 4 days per week and 1 to 2 days per week, respectively. Half 292 (54.5%) of the girls reported that they could use a latrine during break time and 244 (45.5%) reported that they used latrines as needed (Table 5).

### Practices of menstrual hygiene

Nearly all 517 (96.5%) study participants used some type of absorbent material during menstruation. The absorbent materials differed in type; 492 (95.2%) used commercially available sanitary pads, the remaining 25 (4.8%) used homemade absorbents. Among those who used reusable absorbents, 29 (93.5%) used soap and water for cleaning them and 18 (58.1%) dried them in sunlight. About one-thirds 175 (33.9%) changed absorbent material three times a day and 333 (62.1%) did not take a shower daily during menstruation (Table 6).

Responses on the overall practices showed that 289 (53.9%) of the participants practiced good menstrual hygiene (95% CI [49.6, 58.2%]), and the remaining 247 (46.1%) managed poorly (95% CI [41.8, 50.4%]).

**Table 1. Socio-demographic and economic characteristics of high school girls of Dessie City, Amhara Region, northeastern Ethiopia, 27 January to 6 March 2020.**

| Variable | Category | Frequency (*n* = 536) | Percent (%) |
|---|---|---|---|
| **School name** | Hotie | 204 | 38.1 |
| | Kidame Gebeya | 177 | 33.0 |
| | Memhir Akalewold | 76 | 14.2 |
| | Catholic | 66 | 12.3 |
| | Hope Enterprise | 13 | 2.4 |
| **School type** | Public | 457 | 85.3 |
| | Private | 79 | 14.7 |
| **Age (years)** | 13–15 | 239 | 44.6 |
| | 16–19 | 297 | 55.4 |
| **Grade** | 9th | 328 | 61.2 |
| | 10th | 208 | 38.8 |
| **Religion** | Orthodox | 226 | 42.2 |
| | Muslim | 294 | 54.8 |
| | Protestant | 15 | 2.8 |
| | Other* | 1 | 0.2 |
| **Residence** | Urban | 514 | 95.9 |
| | Peri-urban¥ | 22 | 4.1 |
| **Marital status** | Single | 383 | 71.4 |
| | Married | 14 | 2.6 |
| | Divorced | 3 | 0.6 |
| | No response | 136 | 25.4 |
| **Lives with** | Both parents | 389 | 72.6 |
| | Mother only | 67 | 12.5 |
| | Father only | 11 | 2.1 |
| | Relatives | 49 | 9.1 |
| | Alone | 15 | 2.8 |
| | Others** | 5 | 0.9 |
| **Maternal educational status** | Illiterate | 74 | 13.8 |
| | Read and write | 99 | 18.5 |
| | Primary | 106 | 19.8 |
| | Secondary | 139 | 25.9 |
| | College or above | 118 | 22.0 |
| **Paternal educational status** | Illiterate | 33 | 6.2 |
| | Read and write | 85 | 15.8 |
| | Primary | 91 | 17.0 |
| | Secondary | 132 | 24.6 |
| | College or above | 195 | 36.4 |
| **Maternal occupation** | Housewife | 309 | 57.7 |
| | Merchant | 103 | 19.2 |
| | Private organization employee | 34 | 6.3 |
| | Governmental employee | 77 | 14.4 |
| | Daily laborer | 13 | 2.4 |

(*Continued*)

**Table 1.** (Continued)

| Variable | Category | Frequency (n = 536) | Percent (%) |
|---|---|---|---|
| Paternal occupation | Government employee | 182 | 34.0 |
| | Private employee | 63 | 11.7 |
| | Daily laborer | 30 | 5.6 |
| | Self-employed | 189 | 35.3 |
| | Farmer | 65 | 12.1 |
| | Other*** | 7 | 1.3 |
| Receives pocket money from family | Yes | 240 | 44.8 |
| | No | 296 | 55.2 |

*Jehovah's Witnesses Church

**Friends, brothers, sisters

***Religious leader.

¥All 22 students living in peri-urban areas attended school in urban areas of Dessie City. Dessie City administration consists of 6 peri-urban *kebeles* and 10 urban *kebeles*.

## Sanitary pad utilization and related issues

Forty-four (8.2%) out of 536 of the girls did not use commercially made sanitary pads. The main reason given for non-utilization was cost of the commercially available pads (79.5%), followed by their inaccessibility when needed (13.7%), shyness (4.5%), and difficulty disposing of them (2.2%). The majority of the participants 464 (86.6%) asked for money from their families for buying pads and 339 (73.1%) received money from their mothers (Table 7).

## Menstruation and missed school days

Menstrual hygiene-related school absenteeism during the five months preceding the study was reported at 1–2 days and 3–5 days by 57 (79.2%) and 15 (20.8%) girls, respectively. The mean value for missed school days was 1.81 with SD of ±0.98 days. Some of the reasons for absenteeism were dysmenorrhea, fear of staining clothes, and not having sanitary pads (Table 8).

## Factors associated with the practice of menstrual hygiene

In the multivariable analysis, age, grade, maternal education, regularity of menses, knowledge, discussing menstrual hygiene with friends, asking for money from their family for pads were

**Table 2. Gynecological characteristics of the high school girls of Dessie City, Amhara Region, northeastern Ethiopia, 27 January to 6 March 2020.**

| Variable | Category | Frequency (n = 536) | Percent (%) |
|---|---|---|---|
| Age at menarche (years) | < 12 | 43 | 8.0 |
| | 13–15 | 487 | 90.9 |
| | 16–19 | 6 | 1.1 |
| Regularity of menses | Regular | 464 | 86.6 |
| | Irregular | 72 | 13.4 |
| Duration of menses flow | Less than 2 days | 109 | 20.3 |
| | 3 to 7 days | 399 | 74.5 |
| | More than 7 days | 28 | 5.2 |
| Pain during menstruation | Yes | 299 | 55.8 |
| | No | 237 | 44.2 |

**Table 3. Knowledge about menstruation among high school girls of Dessie City, Amhara Region, northeastern Ethiopia, 27 January to 6 March 2020.**

| Variable | Category | Frequency (*n* = 536) | Percent (%) |
|---|---|---|---|
| **Heard about menstruation before menarche** | Yes | 500 | 93.3 |
| | No | 36 | 6.7 |
| **What is menstruation?** | Physiological process | 505 | 94.2 |
| | Pathological process | 4 | 0.8 |
| | Curse from God | 14 | 2.6 |
| | Do not know | 13 | 2.4 |
| **Cause of menstruation** | Hormones | 343 | 64.0 |
| | Curse from God | 43 | 8.0 |
| | Caused by disease | 1 | 0.2 |
| | Other* | 1 | 0.2 |
| | Do not know | 148 | 27.6 |
| **Source of menstrual blood** | Uterus | 304 | 56.7 |
| | Vagina | 42 | 7.8 |
| | Bladder | 3 | 0.6 |
| | Abdomen | 10 | 1.9 |
| | Other** | 8 | 1.5 |
| | Do not know | 169 | 31.5 |
| **Duration of a normal menstrual cycle** | Less than 21 days | 114 | 21.3 |
| | 21 to 35 days | 243 | 45.3 |
| | More than 35 days | 7 | 1.3 |
| | Do not know | 172 | 32.1 |
| **Normal menstrual bleeding duration** | Less than 2 days | 33 | 6.1 |
| | 2 to 7 days | 411 | 76.7 |
| | More than 7 days | 37 | 6.9 |
| | Do not know | 55 | 10.3 |
| **Learned about menstrual hygiene in school** | Yes | 429 | 80.0 |
| | No | 107 | 20.0 |
| **There is foul odor during menstruation** | Yes | 357 | 66.6 |
| | No | 179 | 33.4 |
| **Menstrual blood is unhygienic** | Yes | 366 | 68.3 |
| | No | 170 | 31.7 |
| **Poor hygiene predisposes to infection** | Yes | 459 | 85.6 |
| | No | 77 | 14.4 |
| **Personal hygiene during menstruation reduces pain** | Yes | 380 | 70.9 |
| | No | 156 | 29.1 |
| **Menstruation is a lifelong process** | No | 120 | 22.4 |
| | Yes | 416 | 77.6 |

*Too much water and food

**Heart.

significantly associated with good MHPs. Girls 16–19 years old were 1.9 times more likely to have good MHPs than girls in the 13–15 years age group (AOR = 1.93, 95% CI: [1.22–3.06]). Grade 10 students were 1.9 times more likely to have good MHPs than Grade 9 students [AOR = (1.90, 95% CI: [1.18–3.07]). Girls whose mothers had primary, secondary, or college education were 3.72, 8.54, and 6.78 times more likely to have good MHPs (AOR = 3.72, 95% CI: [1.81–7.63]); AOR = 8.54, 95% CI: (4.18–17.44); AOR = 6.78, 95% CI: [3.28–14.02]), respectively (Table 9).

**Table 4. Source of information and communication about menstruation among high school girls of Dessie City, Amhara Region, northeastern Ethiopia, 27 January to 6 March 2020.**

| Variable | Category | Frequency (*n*) | Percent (%) |
|---|---|---|---|
| **Source of awareness about menarche (*n* = 500)** | Mother | 209 | 41.8 |
| | School (media, teacher) | 106 | 21.2 |
| | Friend | 91 | 18.2 |
| | Elder sister | 76 | 15.2 |
| | Television | 11 | 2.2 |
| | Health professional | 5 | 1.0 |
| | Father | 2 | 0.4 |
| **Discuss menstrual hygiene with friends (*n* = 536)** | Yes | 413 | 77.1 |
| | No | 123 | 22.9 |
| **Communicate about menstruation with your family (*n* = 536)** | Yes | 373 | 69.6 |
| | No | 163 | 30.4 |
| **With whom do you frequently communicate? (*n* = 373)** | Mother | 227 | 60.8 |
| | Father | 6 | 1.6 |
| | Sister | 123 | 33.0 |
| | Another member | 17 | 4.6 |
| **Why no communication about menstruation in your family? (*n* = 163)** | It is shameful | 71 | 43.5 |
| | It is kept as a secret | 80 | 49.1 |
| | Both shameful and kept secret | 7 | 4.3 |
| | Other* | 5 | 3.1 |
| **Communicate about menstruation with teachers (*n* = 536)** | Yes | 182 | 34.0 |
| | No | 354 | 66.0 |

*Nobody cares about it.

The duration between two consecutive menstruation episodes was associated with the practice status. Girls who had regular menses were 1.9 times more likely to have good practices than their irregular counterparts (AOR = 1.85, 95% CI: [1.03–3.32]). High school girls with good knowledge of menses had 2.0 times better menstrual practice than those with poor knowledge (AOR = 2.02, 95% CI: [1.32–3.09]). Students who openly discussed menstrual hygiene with friends were 1.7 times more likely to practice good menstrual hygiene than those who did not discuss it (AOR = 1.79, 95% CI: [1.12–2.86]). Girls who asked for money to purchase pads were two times more likely to practice good menstrual hygiene than those who did not ask (AOR = 2.08, 95% CI: [1.15–3.78]) (Table 9).

## Observational findings

**Water observations.** The main water source at all the schools was piped water in the school yard. Four out of five Schools, the main water sources were not functional during the survey. The water taps in every school were easily accessible even for the youngest children.

**Table 5. Water, sanitation, and hygiene-related issues among high schools of Dessie City, Amhara Region, northeastern Ethiopia, 27 January to 6 March 2020.**

| Variable | Category | Frequency (*n* = 536) | Percent (%) |
|---|---|---|---|
| **Water source functionality in the school** | 5 to 7 days per week | 398 | 74.2 |
| | 2 to 4 days per week | 91 | 17.0 |
| | Fewer than 2 days per week | 47 | 8.8 |
| **When student is allowed to use the latrine** | During breaks only | 292 | 54.5 |
| | Anytime | 244 | 45.5 |

**Table 6. Menstrual hygiene practices among high school girls in Dessie City, Amhara Region, northeastern Ethiopia, 27 January to 6 March 2020.**

| Variable | Category | Frequency (n) | Percent (%) |
|---|---|---|---|
| Use absorbent materials during menstruation (n = 536) | Yes | 517 | 96.5 |
| | No | 19 | 3.5 |
| Absorbent material used during the last 6 months (n = 517) | Commercially made sanitary pads | 492 | 95.2 |
| | Homemade absorbents | 25 | 4.8 |
| Materials used for washing reusable absorbent (n = 31) | Soap and water | 29 | 93.5 |
| | Water only | 2 | 6.5 |
| Drying of washed reusable absorbents (n = 31) | In the sunlight | 18 | 58.1 |
| | In the shade (indoors) | 13 | 41.9 |
| Frequency of changing absorbent material per day (n = 517) | Once | 89 | 17.2 |
| | Twice | 224 | 43.3 |
| | Three times | 175 | 33.9 |
| | More than three times | 29 | 5.6 |
| Cleaning genitalia during menstruation (n = 536) | Yes | 481 | 89.7 |
| | No | 55 | 10.3 |
| Material for cleaning genitals (n = 481) | Soap and water | 169 | 35.1 |
| | Water only | 295 | 61.3 |
| | Paper | 17 | 3.6 |
| Showering daily during menstruation (n = 536) | Yes | 203 | 37.9 |
| | No | 333 | 62.1 |
| Materials used for showering (n = 203) | Soap and water | 167 | 82.3 |
| | Water only | 36 | 17.7 |
| Where do you dispose used menstrual material (n = 517) | Open field | 20 | 3.9 |
| | Latrine | 201 | 38.9 |
| | Waste bin | 295 | 57.0 |
| | Other* | 1 | 0.2 |
| Dispose of pads by wrapping them in paper (n = 517) | Yes | 398 | 77.0 |
| | No | 119 | 23.0 |

*Burying.

**Table 7. Sanitary pad-related issues among high -school girls of Dessie City, Amhara Region, northeastern Ethiopia, 27 January to 6 March 2020.**

| Variable | Category | Frequency (n) | Percent (%) |
|---|---|---|---|
| Reason given for non-utilization of commercial sanitary pads (n = 44) | Cost | 35 | 79.6 |
| | Not available | 6 | 13.7 |
| | Difficulty in disposal | 1 | 2.2 |
| | Shyness | 2 | 4.5 |
| Asked for money from family for pad (n = 536) | Yes | 464 | 86.6 |
| | No | 72 | 13.4 |
| From whom do you get money for buying pads? (n = 464) | Mother | 339 | 73.1 |
| | Father | 56 | 12.1 |
| | Elder sister | 51 | 11.0 |
| | Brother | 16 | 3.4 |
| | Other* | 2 | 0.4 |

*Niece.

**Table 8. Reasons for menstruation-related school absenteeism among high school girls of Dessie City, Amhara Region, northeastern Ethiopia, 27 January to 6 March 2020.**

| Reason (N = 72)[*], [#] | Frequency (n) | Percent (%) |
|---|---|---|
| Afraid of staining my clothes | 17 | 23.6 |
| Afraid of others making fun of me | 11 | 15.3 |
| A period can cause pain | 31 | 43.1 |
| Periods can make me feel uncomfortable | 17 | 23.6 |
| There is no place for girls to wash | 9 | 12.5 |
| There is no disposal system for pads | 1 | 1.4 |
| I do not have sanitary pads | 9 | 12.5 |
| There is no place for girls to change pads | 7 | 9.7 |

[*]Out of 72 girls who have been reported school absenteeism, there were multiple responses for the reasons of menstruation-related school absenteeism. Thus, the sum of the percentage for the reasons was more than 100%.
[#]Fifty seven (79.2%) and 15 (20.8%) out of 72 girls reported 1–2 days and 3–5 days school absenteeism due to menstrual hygiene during the five months preceding the study, respectively.

**Sanitation observations.** All five schools had gender-based toilets on their compounds. In terms of functionality, most toilets could be used, but there were problems with the physical infrastructure such as deterioration of concrete, missing doors, and deteriorating roofs. Thus, most of the toilets were partially functional and some were not functional. There was smell, signs of fecal matter and urine in most toilets, and some toilets had a strong smell and more visible signs of excreta. These toilets were classified as either somewhat clean or not clean. The interior of the toilets in most schools was fairly dark. If girls were able to look at their uniforms and tell if there was a bloodstain, those toilets were classified as being somewhat dark. Some toilets were too dark for girls to see if their uniforms were stained. None of the toilets had doors lockable from the inside and some had no door at all. There was a waste bin in most toilets but no anal cleaning materials were available. None of the toilets were accessible for disabled persons.

**Hygiene observations.** There were hand-washing facilities in all five schools. The locations of hand-washing facilities were a long way from the toilet blocks. Water and soap or ash were not available in any of the five schools during the observation period. Sanitary napkins were not available for emergency/accidental situations in any governmental school but were made available in the private school when needed. None of the schools had private facilities where girls could bathe or change sanitary pads.

## Discussion

This school-based cross-sectional study examined selected socio-demographic, and gynecological variables, knowledge about menstruation, sources of information, communication, sanitary pad-related and WASH variables about MHPs. The study assessed whether menstrual hygiene was associated with the aforementioned factors. Our findings show that 53.9% (95% CI: 49.6, 58.2%) of the schoolgirls practiced good menstrual hygiene. Age 16–19 years, grade level of 10, maternal education (primary, secondary, and college), having regular menses, good knowledge, discussing menstrual hygiene with friends, and asking family for money for pads were significantly associated with good MHPs. Observational findings revealed that none of the five high schools had girl-friendly WASH facilities.

The prevalence of good MHPs in Dessie is similar to the 57.0% reported by a study in Adama City in Ethiopia [26], the (50.8%) reported in Ghana [32], and the 47.5% reported in

**Table 9. Bivariate and multivariable logistic regression analysis for factors associated with menstrual hygienic practices among high school girls in Dessie City, northeastern Ethiopia, 27 January to 6 March 2020.**

| Variable | Menstrual hygiene practice status | | COR (95% CI) | AOR (95% CI) |
|---|---|---|---|---|
| | Good (N = 289) | Poor (N = 247) | | |
| | *n* (%) | *n* (%) | | |
| **Age (years)** | | | | |
| 13–15 | 104(36.0) | 135(54.7) | Ref | Ref |
| 16–19 | 185(64.0) | 112(45.3) | 2.14(1.50–3.01) | 1.93(1.22–3.06) |
| **Grade** | | | | |
| 9th | 149(51.6) | 179(72.5) | Ref | Ref |
| 10th | 140(48.4) | 68(27.5) | 2.47(1.72–3.55) | 1.90(1.18–3.07) |
| **Marital status** | | | | |
| Single | 207(71.6) | 176(71.3) | Ref | Ref |
| Married | 7(2.4) | 10(4.0) | 0.59(0.16–1.44) | 0.38(0.10–1.46) |
| Not applicable | 75(26.0) | 61(24.7) | 1.04(0.71–1.55) | 1.02(0.65–1.61) |
| **Live with** | | | | |
| Both parents | 218(75.4) | 171(69.2) | Ref | Ref |
| Mother only | 31(10.7) | 36(14.6) | 0.68(0.40–1.14) | 0.58(0.32–1.04) |
| Father only | 3(1.0) | 8(3.2) | 0.29(0.08–1.13) | 0.51(0.11–2.35) |
| Relatives | 23(8.0) | 26(10.5) | 0.69(0.38–1.26) | 0.96(0.48–1.93) |
| Alone | 14(4.8) | 6(2.4) | 2.16(0.68–6.89) | 1.28(0.34–4.85) |
| **Maternal education** | | | | |
| Illiterate | 16(5.5) | 58(23.5) | Ref | Ref |
| Read and write | 39(13.5) | 60(24.3) | 2.36(1.19–4.67) | 1.89 (0.91–3.93) |
| Primary | 57(19.7) | 49(19.8) | 4.22(2.15–8.26) | 3.72 (1.81–7.63) |
| Secondary | 97(33.6) | 42(17.0) | 8.37(4.32–16.22) | 8.54(4.18–17.44) |
| College | 80(27.7) | 38(15.4) | 7.63(3.89–14.99) | 6.78 (3.28–14.02) |
| **Paternal education** | | | | |
| Illiterate | 9(3.1) | 24(9.7) | Ref | Ref |
| Read and write | 39(13.5) | 46(18.6) | 2.26(0.94–5.43) | 1.28(0.45–3.66) |
| Primary | 43(14.9) | 48(19.4) | 2.39(1.00–5.70) | 0.91(0.32–2.58) |
| Secondary | 82(28.4) | 50(20.3) | 4.37(1.88–10.16) | 0.99(0.35–2.79) |
| College | 116(40.1) | 79(32.0) | 3.92(1.73–8.87) | 0.71(0.25–2.01) |
| **Maternal occupation** | | | | |
| Housewife | 154(53.3) | 155(62.8) | Ref | Ref |
| Merchant | 56(19.4) | 47(19.0) | 1.20(0.77–1.88) | 1.06(0.61–1.82) |
| Private organization employee | 18(6.2) | 16(6.5) | 1.13(0.56–2.30) | 0.67(0.29–1.57) |
| Governmental employee | 55(19.0) | 22(8.9) | 2.52(1.46–4.33) | 1.18(0.58–2.39) |
| Daily laborer | 6(2.1) | 7(2.8) | 0.86(0.28–2.63) | 1.36(0.33–5.67) |
| **Paternal occupation** | | | | |
| Government employee | 102(35.3) | 80(32.4) | Ref | Ref |
| Private employee | 39(13.5) | 24(9.7) | 1.28(0.71–2.29) | 1.30(0.63–2.67) |
| Daily laborer | 13(4.5) | 17(6.9) | 0.60(0.28–1.31) | 0.96(0.35–2.68) |
| Self-employed | 105(36.3) | 84(34.0) | 0.98(0.65–1.48) | 1.03(0.60–1.77) |
| Farmer | 28(9.7) | 37(15.0) | 0.59(0.34–1.05) | 1.00(0.44–2.27) |
| Other | 2(0.7) | 5(2.0) | 0.31(0.06–1.66) | 0.34(0.04–2.64) |
| **Regular menses** | | | | |
| No | 31(10.7) | 41(16.6) | Ref | Ref |
| Yes | 258(89.3) | 206(83.4) | 1.66(1.00–2.73) | 1.85(1.03–3.32) |

(*Continued*)

**Table 9.** (Continued)

| Variable | Menstrual hygiene practice status | | COR (95% CI) | AOR (95% CI) |
|---|---|---|---|---|
| | **Good (N = 289)** | **Poor (N = 247)** | | |
| | **n (%)** | **n (%)** | | |
| **Duration of menses flow** | | | | |
| < 2 days | 60(20.8) | 49(19.8) | 1.89(0.81–4.42) | 1.79(0.64–5.01) |
| 3 to 7 days | 218(75.4) | 181(73.3) | 1.86(0.85–4.08) | 1.54(0.58–4.09) |
| >7 days | 11(3.8) | 17(6.9) | Ref | Ref |
| **Knowledge status** | | | | |
| Poor | 68(23.5) | 102(41.3) | Ref | Ref |
| Good | 221(76.5) | 145(58.7) | 2.29(1.58–3.32) | 2.02 (1.32–3.09) |
| **Discuss menstrual hygiene with friends** | | | | |
| No | 52(18.0) | 71(28.7) | Ref | Ref |
| Yes | 237(82.0) | 176(71.3) | 1.84(1.22–2.76) | 1.79 (1.12–2.86) |
| **Communicate about menstruation with family** | | | | |
| No | 77(26.6) | 86(34.8) | Ref | Ref |
| Yes | 212(73.4) | 161(65.2) | 1.47(1.02–2.13) | 1.19(0.76–1.85) |
| **Water source functionality in the school** | | | | |
| Fewer than 2 days per week | 18(6.2) | 29(11.7) | Ref | Ref |
| 2 to 4 days per week | 49(17.0) | 42(17.0) | 1.88(0.92–3.85) | 1.41(0.61–3.23) |
| 5 to 7 days per week | 222(76.8) | 176(71.3) | 2.03(1.09–3.78) | 1.62(0.78–3.36) |
| **Ask for money for pads from family** | | | | |
| No | 24(8.3) | 48(19.4) | Ref | Ref |
| Yes | 265(91.7) | 199(80.6) | 2.66(1.58–4.50) | 2.08 (1.15–3.78) |

COR, crude odds ratio; AOR, adjusted odds ratio; CI, confidence interval; Ref, reference category.

West Bengal (India) [33]. Lower rates (39.7% and 35.4%) were reported in southern Ethiopia [27] and in Habru Town, Ethiopia [21], respectively. Possible reasons for the low number in Habru could be the measurement used to assess the practice status (a single question: sanitary pad utilization), a difference in the study period, and the fact that most of the study participants were from rural areas. The southern Ethiopia study had data discrepancies, mainly due to the participants' poor knowledge of menstrual hygiene compared to the knowledge level in the current study.

A study in Egypt found that 90% of students had acceptable MHPs [34]. The reason for this high rate may be the relatively high socioeconomic level of participants, good WASH facilities, and the presence of bathing facilities in this industrial and agricultural community, all of which facilitated good MHPs. In Dessie, by contrast, our observations showed that schools had poor WASH services, toilets lacked doors and locks, and none of the schools had facilities for bathing. This situation was also found in Tanzania, where the lack of soap, hand washing facilities, emergency pads, or privacy were important determinants of poor MHPs [35].

The rate of changing pads with adequate frequency among girls in this study was similar with Ghanaian and West Bengal (India) studies, where the proportions were 45.2% and 42.3%, respectively [32, 33]. This low frequency may emanate from the high cost of pads, unawareness of the need to change pads frequently, failure of the schools to provide sanitary pads at least in emergencies, and lack of changing rooms. Higher rates (51.9% and 62.4%) of satisfactorily changing pads were reported by western and southern Ethiopian studies, respectively [27, 28]. These discrepancies may be due to the fact that most participants in some of these studies used

homemade absorbents that were affordable and therefore more likely to be changed frequently than the commercial pads that were used by most girls in Dessie schools.

Among those who used reusable absorbents, 41.9% dried them without sunlight. This rate was comparable with those found in other Ethiopian studies, and the explanation could be that girls did not want to be seen handling absorbents outside [27, 28]. In a West Bengal (India) study, 72.2% of schoolgirls dried the reusable cloth without sunlight because of the underprivileged status of West Bengal (India) adolescents and their fear of being seen in public drying their reusable pads [33]. Another study in India indicated that rural participants dried their sanitary pads inside their houses because menstruation is considered impure and dirty, something that should be hidden due to taboos in that society [16, 36]. Washing reusable absorbents and drying them in sunlight may be a sustainable sanitary option because the sun is a natural sterilizer. To avoid contamination, the materials need to be stored in a clean, dry place for reuse.

In the current study, only 35.1% of participants used soap and water to clean their genitalia, a percentage lower than in similar studies in southern Ethiopia and West Bengal (India) [27, 33]. The discrepancy may be due to the unavailability of soap on school compounds and also lack of awareness in the current study. The present study showed that only 37.9% of participants took a daily shower during menstruation. Studies in southern Ethiopia and Ghana reported 56.4% and 94.4% daily showering, respectively [27, 32]. These discrepancies are possibly due to the warmer climate in Ghana than in Dessie and the requirement for frequent bathing among Muslims in Ghana, where almost all study participants were Muslims. Inadequate availability of water, lack of soap, and lack of showers for girls in the school compounds, as observed during our survey, may be additional reasons.

The findings showed that 57.1% of participants disposed of used menstrual materials in waste bins. Similar results were reported by other studies: 44.7% in Ethiopia and 43.3% in Nigeria [27, 37]. A descriptive study in Zambia showed that girls preferred to dispose used menstrual materials in pit latrines rather than waste bins for fear that they could be retrieved for witchcraft against them [38]. Improper disposal of pads can increase solid waste, and the practice of not wrapping the absorbent materials and disposing them in the toilet is unsightly and may create breeding places for insects and vermin, leading to the spread of disease [32, 37]. Our observations at one of the schools found that some used pads that were not wrapped in paper in waste bins.

Identifying factors associated with good MHPs was the second objective of the study. The prevalence of good menstrual hygiene in the 16-to-18 age group was higher than in the 13-to-15 age group. Similar associations of age and practices were reported from southern Ethiopia and Nigeria [27, 37]. Older girls have had more opportunities to obtain relevant information about menstrual hygiene and practicing safe hygiene during menstruation than younger girls [27]. Also, girls in our study had more experience regarding menarche and menstruation management compared to their counterparts.

In this study, Grade 10 students practiced better menstrual hygiene than Grade 9 students. This association is also supported by studies in Oromia Region in Ethiopia and in Indonesia [31, 39]. The possible explanation might be that older students can increase their awareness of menstruation and proper MHP through the school curriculum and informal communication among classmates.

Similarly, maternal educational status, with an academic completion of primary, secondary, or college levels, was associated with good menstrual hygiene. Various studies worldwide noted this association, including other studies in Ethiopia, Nigeria, and West Bengal (India) [16, 21, 28, 33, 37]. The reason may be that educated mothers are more familiar with good

MHPs, are more willing to discuss menstruation with their daughters, provide sanitary pads, and insist that girls clean their genitalia during menstruation.

This study further discovered that study participants with a regular menstruation cycle had better MHPs than their irregular counterparts. The explanation may be that girls with irregular menses cannot anticipate their onset, and may therefore be less prepared for proper menstrual hygiene (i.e., they may not buy or obtain sanitary pads in time). The unpredictability of their menstruation may affect their psychological and emotional states, diminishing their motivation and commitment to engaging in good hygienic practices. We found no studies supporting this finding, but a study conducted in Bahir Dar University, Ethiopia showed an association of irregular menstruation with premenstrual syndrome [40]. This finding suggests that the regularity of menses associated with premenstrual syndrome, which affects a girl's emotions, physical health, and behavior during certain days of the menstrual cycle before the onset of her menses and, in turn, influences her hygienic practices during menstruation.

Another finding of this study was that knowledge about menstruation helps girls maintain good menstrual hygiene. Study subjects with a good level of knowledge regarding menstruation and menstrual hygiene practiced safer menstrual hygiene than their counterparts. Studies conducted in southern Ethiopia, Nigeria and Indonesia also found that knowledge about menstruation was significantly associated with good MHPs [27, 31, 41]. Thus girls should have sufficient knowledge surrounding menstruation, the menstrual cycle, and menstrual hygiene even before menarche. The collective knowledge of age at menarche, menstrual cycle, and duration of menstrual flow in adolescents is useful for allaying fears and psychological trauma that may arise from an unexpected appearance of blood at menarche. Besides, sufficient knowledge of menstruation is expected to empower adolescents to distinguish between physiologic and abnormal uterine bleeding [16, 41]. Overall, better knowledge about menses and menstrual hygiene helps girls accept the natural phenomenon as a normal physiological process and follow proper hygienic practices.

Discussing menstrual hygiene with friends was another significant factor in this study. Study subjects who discussed menstrual hygiene with friends practiced safer menstrual hygiene than those who did not. An Indian study indicated that women who openly discussed menses experiences enhanced ones understanding of menstruation [16]. These discussions can raise the level of knowledge about menstrual hygiene, inform where to borrow sanitary pads when needed, enable them to share experiences and receive emotional support, decrease psychological stress, and boost confidence. Information from peers is simpler to obtain and process because peer conversations about a sensitive subject take place in a casual atmosphere that encourages girls to share their concerns without apprehension.

Moreover, good MHPs of the participants were associated with receiving a regular allowance from the family to purchase pads. Girls who received a regular allowance for sanitary products had better menstrual hygiene management than those who did not receive such an allowance. This finding is supported by studies in western Ethiopia and Ghana [29, 32]. The explanation for this may be that girls who receive money whenever they need it for purchasing sanitary pads regularly utilize sanitary pads, leading to safer practices of menstrual hygiene. Otherwise, they are forced to ask peers for pads [42].

## Limitations of the study

Since the study used a cross-sectional study design, it is difficult to build up causal relationships between the outcome and exposure variables. Further studies are encouraged using randomized control trial to address this gap and also to identify socially acceptable, sustainable, affordable, and environmentally friendly sanitary pads. Our study used only quantitative data

collection and analysis, and it is not triangulated with qualitative evidence. During data collection, using close–ended questions required study participants to select among the listed options in most questions, which may limit further options and more nuanced evaluation. To address this limitation, we explored different previous qualitative and quantitative studies and tried to include all possible options for each question.

## Conclusions

We found that more than half of high school girls had good menstrual hygiene practices. Factors significantly associated with good menstrual hygiene practices include age (16–19 years), grade level, maternal education, regular menses, good knowledge regarding menstruation, discussing menstrual hygiene with friends and obtaining money for pads from the family. Schools should take the pivotal role towards the awareness creation of safe MHP. In addition, educating high school student's mothers should be a prioritized intervention area to eliminate the problem of menstrual hygiene among daughters. Furthermore, in order to improve the MHP among high school girls, further attention is needed for increasing knowledge regarding menstruation among high school girls, encouraging high school girls' family to support by buying sanitary pads and promoting discussion among friends about menstrual hygiene. This should include educating mothers about safe MHPs and encouraging them to teach their daughters about menstrual hygiene. Implementation of girl-friendly WASH services in all schools should be prioritized by programmers, managers, concerned stakeholders (governmental and non-governmental organizations) and policy makers.

## Supporting information

**S1 Appendix. English version of the questionnaire.** Survey of menstrual hygiene practices among high school girls in urban areas in northeastern Ethiopia: A neglected issue in water, sanitation, and hygiene research.
(DOCX)

**S2 Appendix. Amharic (local language) version of the questionnaire.** Survey of menstrual hygiene practices among high school girls in urban areas in northeastern Ethiopia: A neglected issue in water, sanitation, and hygiene research.
(DOCX)

**S1 Dataset. Minimal data for survey of menstrual hygiene practices among high school girls in urban areas in northeastern Ethiopia: A neglected issue in water, sanitation, and hygiene research.**
(XLSX)

## Acknowledgments

We would like to express our gratitude to Dessie City Education Department Office for providing information. We thank all participating high school directors, girls' club coordinators, and unit leaders for facilitating the data collection process. We are also grateful to the data collectors and study participants for their assistance and cooperation. Last but not least, we also thank Lisa Penttila for language editing of the manuscript.

## Author Contributions

**Conceptualization:** Yohannes Habtegiorgis, Mastewal Arefaynie, Mistir Lingerew, Metadel Adane.

**Data curation:** Yohannes Habtegiorgis, Metadel Adane.

**Formal analysis:** Yohannes Habtegiorgis, Metadel Adane.

**Funding acquisition:** Yohannes Habtegiorgis.

**Investigation:** Yohannes Habtegiorgis, Tadesse Sisay, Asmamaw Malede, Melaku Yalew, Mastewal Arefaynie, Yitayish Damtie, Bereket Kefale, Tesfaye Birhane Tegegne, Elsabeth Addisu, Mistir Lingerew, Leykun Berhanu, Gete Berihun, Tarikuwa Natnael, Masresha Abebe, Alelgne Feleke, Adinew Gizeyatu, Ayechew Ademas, Zinabu Fentaw, Tilaye Matebe Yayeh, Metadel Adane.

**Methodology:** Yohannes Habtegiorgis, Tadesse Sisay, Asmamaw Malede, Melaku Yalew, Mastewal Arefaynie, Yitayish Damtie, Bereket Kefale, Tesfaye Birhane Tegegne, Elsabeth Addisu, Mistir Lingerew, Leykun Berhanu, Gete Berihun, Tarikuwa Natnael, Masresha Abebe, Alelgne Feleke, Adinew Gizeyatu, Ayechew Ademas, Zinabu Fentaw, Tilaye Matebe Yayeh, Fitsum Dangura, Metadel Adane.

**Project administration:** Yohannes Habtegiorgis, Asmamaw Malede, Tesfaye Birhane Tegegne, Elsabeth Addisu, Adinew Gizeyatu, Ayechew Ademas, Zinabu Fentaw, Tilaye Matebe Yayeh, Fitsum Dangura, Metadel Adane.

**Resources:** Yohannes Habtegiorgis, Tadesse Sisay, Helmut Kloos, Asmamaw Malede, Melaku Yalew, Mastewal Arefaynie, Yitayish Damtie, Bereket Kefale, Tesfaye Birhane Tegegne, Elsabeth Addisu, Mistir Lingerew, Leykun Berhanu, Gete Berihun, Tarikuwa Natnael, Masresha Abebe, Alelgne Feleke, Adinew Gizeyatu, Ayechew Ademas, Zinabu Fentaw, Tilaye Matebe Yayeh, Fitsum Dangura, Metadel Adane.

**Software:** Yohannes Habtegiorgis, Tadesse Sisay, Helmut Kloos, Asmamaw Malede, Melaku Yalew, Mastewal Arefaynie, Yitayish Damtie, Bereket Kefale, Tesfaye Birhane Tegegne, Elsabeth Addisu, Mistir Lingerew, Leykun Berhanu, Gete Berihun, Tarikuwa Natnael, Masresha Abebe, Alelgne Feleke, Adinew Gizeyatu, Ayechew Ademas, Zinabu Fentaw, Tilaye Matebe Yayeh, Metadel Adane.

**Supervision:** Yohannes Habtegiorgis, Tadesse Sisay, Asmamaw Malede, Melaku Yalew, Mastewal Arefaynie, Yitayish Damtie, Bereket Kefale, Tesfaye Birhane Tegegne, Elsabeth Addisu, Mistir Lingerew, Leykun Berhanu, Gete Berihun, Tarikuwa Natnael, Masresha Abebe, Alelgne Feleke, Adinew Gizeyatu, Ayechew Ademas, Zinabu Fentaw, Tilaye Matebe Yayeh, Fitsum Dangura, Metadel Adane.

**Validation:** Yohannes Habtegiorgis, Tadesse Sisay, Helmut Kloos, Asmamaw Malede, Melaku Yalew, Mastewal Arefaynie, Yitayish Damtie, Bereket Kefale, Tesfaye Birhane Tegegne, Elsabeth Addisu, Mistir Lingerew, Leykun Berhanu, Gete Berihun, Tarikuwa Natnael, Masresha Abebe, Alelgne Feleke, Adinew Gizeyatu, Ayechew Ademas, Zinabu Fentaw, Tilaye Matebe Yayeh, Fitsum Dangura, Metadel Adane.

**Visualization:** Yohannes Habtegiorgis, Tadesse Sisay, Helmut Kloos, Asmamaw Malede, Melaku Yalew, Mastewal Arefaynie, Yitayish Damtie, Bereket Kefale, Tesfaye Birhane Tegegne, Elsabeth Addisu, Mistir Lingerew, Leykun Berhanu, Gete Berihun, Tarikuwa Natnael, Masresha Abebe, Alelgne Feleke, Adinew Gizeyatu, Ayechew Ademas, Zinabu Fentaw, Tilaye Matebe Yayeh, Fitsum Dangura, Metadel Adane.

**Writing – original draft:** Yohannes Habtegiorgis, Metadel Adane.

**Writing – review & editing:** Helmut Kloos, Metadel Adane.

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
