## [Decision Letter · Decision Letter 0]

19 Jan 2021

PONE-D-20-39090

Menstrual Hygiene Practices among High School Girls in Urban Areas in Northeastern Ethiopia: A Neglected Issue in Water, Sanitation, and Hygiene Research

PLOS ONE

Dear Dr. Adane (PhD),

Thank you for submitting your manuscript to PLOS ONE. After careful consideration, we feel that it has merit but does not fully meet PLOS ONE’s publication criteria as it currently stands. Therefore, we invite you to submit a revised version of the manuscript that addresses the points raised during the review process.

Considering the reviewer comments and my own reading of the paper, I am going with a decision of major revision.  The paper needs to improve the Introduction and Background. Specifically, it needs to bring what the study is adding to the literature .  Explain the design of the study carefully addressing the reviewer 1 comments.  Refer to following articles while revising the paper and also to compare your results with other developing countries to enrich the discussion. 

MALHOTRA A, GOLI S, COATES S, MOSQUERA-VASQUEZ MA. Factors associated with knowledge, attitudes, and hygiene practices during menstruation among adolescent girls in Uttar Pradesh. waterlines. 2016 Jul 1:277-305.

Goli S, Sharif N, Paul S, Salve PS. Geographical disparity and socio-demographic correlates of menstrual absorbent use in India: A cross-sectional study of girls aged 15–24 years. Children and Youth Services Review. 2020 Oct 1;117:105283.

We look forward to receiving your revised manuscript.

Kind regards,

Srinivas Goli, Ph.D.

Academic Editor

PLOS ONE

Additional Editor Comments:

Considering the reviewer comments and my own reading of the paper, I am going with a decision of major revision. The paper needs to improve the Introduction and Background. Specifically, it needs to bring what the study is adding to the literature . Explain the design of the study carefully addressing the reviewer 1 comments. Refer to following articles while revising the paper and also to compare your results with other developing countries to enrich the discussion.

MALHOTRA A, GOLI S, COATES S, MOSQUERA-VASQUEZ MA. Factors associated with knowledge, attitudes, and hygiene practices during menstruation among adolescent girls in Uttar Pradesh. waterlines. 2016 Jul 1:277-305.

Goli S, Sharif N, Paul S, Salve PS. Geographical disparity and socio-demographic correlates of menstrual absorbent use in India: A cross-sectional study of girls aged 15–24 years. Children and Youth Services Review. 2020 Oct 1;117:105283.

Journal Requirements:

2. Please include additional information regarding the survey or questionnaire used in the study and ensure that you have provided sufficient details that others could replicate the analyses. For instance, if you developed a questionnaire as part of this study and it is not under a copyright more restrictive than CC-BY, please include a copy, in both the original language and English, as Supporting Information.  If the original language is written in non-Latin characters, for example Amharic, Chinese, or Korean, please use a file format that ensures these characters are visible.

3. You indicated that you had ethical approval for your study. In your Methods section, please ensure you have also stated whether you obtained consent from parents or guardians of the minors (<18 years old) included in the study or whether the research ethics committee or IRB specifically waived the need for their consent.

Reviewers' comments:

Reviewer's Responses to Questions

**Comments to the Author**

1. Is the manuscript technically sound, and do the data support the conclusions?

Reviewer #1: No

Reviewer #2: No

Reviewer #3: Yes

2. Has the statistical analysis been performed appropriately and rigorously? 

Reviewer #1: No

Reviewer #2: No

Reviewer #3: Yes

3. Have the authors made all data underlying the findings in their manuscript fully available?

Reviewer #1: No

Reviewer #2: No

Reviewer #3: Yes

4. Is the manuscript presented in an intelligible fashion and written in standard English?

Reviewer #1: Yes

Reviewer #2: Yes

Reviewer #3: Yes

5. Review Comments to the Author

Reviewer #1: The authors attempted to explore menstrual hygiene practices among high school girls in an urban setting of North-eastern Ethiopia. The study has several issues with the research design. The research design is not appropriate and, therefore, needs significant revision. How a study conducted in the urban area can have sample from rural areas. The authors have chosen five schools; however, the sample size from school varies drastically. What was the basis for choosing respondents from various schools? The formation of certain variables is not up to the mark. The close-ended questions that have limited options have the potential to include other options to choose from—thus limiting the study potential. The study failed to write a proper conclusion section. The study is full of limitations yet failed to include limitations in the text. The sampling procedure is confusing. The authors failed to write about sampling correctly. The study failed to impress, as it has a poor research design. Maybe the research design is not so poor- possibilities that authors failed to describe it adequately—suggested that the author describe the research design adequately. It is not clear whether the consent was written or verbal as information related to both modes of consent is given in text separately. Further, how distributing sanitary pads affected the response can be discussed. The article is written in standard English, however, failed to impress with its study design.

Reviewer #2: I have read the manuscript with interest. This article is interesting; however, it lacks conceptualization. I have some suggestion to make the article a better version.

Abstract:

1. Abstract seems clumsy. Background can be improved. The authors failed to build a proper rationale in the background section. The third line of the abstract shall not start with ‘but.’ Consider rewriting it.

2. The heading in the abstract – Methods, results, and Conclusion must be assigned to a different line. Currently, the heading is placed at the end of the sentence.

3. Why menstrual hygiene practice is higher among girls whose mothers had secondary level education than those whose mothers had only primary level education; when menstrual hygiene practice is lower among girls whose mothers had college-level education than those whose mothers had secondary level education. Need to elaborate on this finding.

4. The conclusion is erratic. Implications are not concrete. Moreover, keeping point number 3 cited above in mind- what level of education among mothers may be significant, as discussed in the study conclusion?

Declaration Section:

5. In the ethics statement, the authors stated that 90 disposable sanitary pads were distributed to those individuals who were menstruating and could not afford them. Did the authors ask the respondents about their ongoing menstruation? If yes, how authors concluded that they could not afford the sanitary pads? Further, what implications this may have on the study design as it is clear from the beginning that such respondents were poor (Since they cannot afford sanitary pads)? Moreover, the distribution of sanitary pads not influencing the respondents’ decision to participate in the study.

6. Authors stated that all relevant data are within the manuscript and its supporting files. However, I could not find any relevant data associated with this submission.

Manuscript:

7. Line number 94-95; the reference in the study context is a bit old. Try citing the latest reference and current prevalence.

8. Authors have used ‘good menstrual hygiene practices’ and ‘menstrual hygiene practices’ interchangeably, which is confusing.

9. Introduction is nicely articulated; however, it needs a bit of editing for ease of flow.

10. Authors have mentioned in the abstract as well as in the manuscript text that the questionnaire was pre-tested. However, the information regarding the same is not given in full. When was it tested? Under whom guidance it was tested? The population upon which it was tested? - and other such details are missing.

11. Did authors use 5 percent of margin of error conveniently or reached after consensus among various authors or some other relevant authority?

12. The study was conducted in an urban area of Dessie city. However, to calculate the sample size, the prevalence used was that of Adama town. It may be highly likely that menstrual hygiene practices' prevalence may be contrasting in these two cities.

13. Representation of ‘z’ in the sample size formula needs to be written in full for a general understanding of the readers.

14. How do authors reach the distribution of sample size between public (467) and private (79) schools? It is stated that distribution was proportionally done? Please elaborate on the proportion used.

15. The sampling procedure is confusing. Consider presenting it through a flow chart. The authors stated that it was a two-stage sampling procedure; however, it is confusing in detailed text.

16. There is a contradiction in the statement by the authors. Somewhere, written consent was used, and at some places, verbal consent (Line- 199) was used in the text. Was the consent written or verbal or a mix of both- clarify. If verbal, how does the authenticity of the same can be produced?

17. It is suggested to give the number of field investigators used for data collection. Were the same set of investigators used for all the schools, or different schools were covered by different investigators? Data collection was carried out simultaneously in various schools, or was it collected one after the other?

18. Direct observations were made to assess the suitability of the WASH facilities (Line- 201-202). How do investigators conclude the suitability of WASH facilities?

19. Line number 206-207. Unpublished and published research was reviewed. Kindly elaborate on what kind of unpublished literature?

20. It is suggested to give the equation for the statistics used in the study.

21. The study design is having severe issues. The number of respondents varies from 13 students in a school to 204 students from other school. Moreover, students were chosen from only two classes- 9th and 10th standard- Then, under what circumstances, it is possible that the age of the respondents varies from 13-19 years. It will be better if age-wise classification is presented in table 1.

22. Study title clearly state that the study is conducted in urban area. Also, the methodology states the same. Then how is it possible to have a rural sample in the study? Are these respondents traveling from rural areas to urban schools?

23. Authors stated that the response rate was around 98 percent. The study says that 98 percent of the respondents responded to all the questions. If so, how can 25 percent of the respondents have not recorded their answer (no response) for the marital status category? Is it like 98 percent agreed to record their responses and then left unanswered questions leading to an incomplete questionnaire?

24. Table 3- option related to What is menstruation and other such questions may also have various other options than provided in the questionnaire. The questions asked might have included other categories also.

25. Table 4- why sample is different for each variable?

26. Table 4- Source of awareness about menarche- the total adds to 500, and it was stated that n is 500. I wonder that the sources cannot be single for such information. Girls may receive information about menarche from both parents as well as their friends simultaneously and also from the media. Single source of information is undermining the study potential.

27. Table 4- With whom do you communicate frequently?- The question is confusing- communication for what?

28. Table 5- water source functionality in the school- It is confusing how water functionality can be fewer than two days or 2-4 days in a school. The water functionality might be regular, and only in case of some service disruption, the water functionality in the school may be an issue.

29. Table 5- When student is allowed to use latrine- The question has limited options. It is not possible that every time a respondent may be allowed to go to the latrine either in the break. It may also depend upon the ongoing situation/lecture importance in school. Sometimes, teachers may or may not allow students to go to latrine. So, the options are not correct.

30. Table 6- result found that 517 (96.5%) of the respondent use absorbent material. Moreover, all of these respondents change their absorbent at least once daily. At other places, the authors noted that they distributed certain sanitary pads as respondents could not afford one. How can this be the case when 96 percent of the respondent are using and changing their absorbents daily?

31. In the odds ratio table 9- the age-group can be divided into two groups only; 13-15 and remaining.

32. What is not applicable category in marital status?

33. Why father’s occupation is not significant in the odds ratio in table 9. It is understood that if a father is working, it is more likely that a girl may afford sanitary napkin. Elaborate on this finding in the context of this study.

34. Water source functionality in the school is not significant in the odds ratio model- elaborate on this finding.

35. The authors shall include the strength and limitation of this study.

36. Conclusions are erratic. The study did not include the disabled; however, in the conclusion section, the authors propose policies for the disabled.

37. It Seems that conclusion is not relevant to the study objective and just written haphazardly after going through certain available literature.

38. Please define what is environmental friendly sanitary napkin as discussed in conclusion section and why authors are proposing that special attention is to be given to lower grade levels students when the study population include 9th and 10th grade students?

39. On the other hand, the use of drugs for irregular menses, especially contraceptive pills, may currently not be feasible in Ethiopia due to negative cultural attitudes of parents toward their use- How does this sentence used in conclusion section is relevant to the study context?

Reviewer #3: The paper on “Menstrual Hygiene Practices among High School Girls in Urban Areas in Northeastern Ethiopia: A Neglected Issue in Water, Sanitation, and Hygiene Research” is well written paper by the respective authors. The topic is of the prime importance in the field of public health domain and can be published after minor revisions.

Abstract

The abstract is well written

Introduction

Overall the introduction is well-written and covered all domains that need to be highlighted in the Introduction. I request authors to please write hypothesis in the last para of the Introduction.

Methods and materials

The section is very well explained and detailed write-up is provided for the same.

Results

Well written.

Please do not highlight the text in page -18 line 357.

Discussion

All the findings are well discussed in the respective section.

Just a small suggestion where ever “West Bengal” is being written, please write in brackets (India) as far as I know it’s the same region which is from India.

I request the authors to write one para (3-4 lines) as limitations and strength of the study. Please write that in the end of the discussion .

References

Some references do not have publisher’s name. Please do amend those references.

Tables

Tables are well structured.

The highlighted estimates are not needed. I mean do not highlight anything in the tables.

COR and AOR full forms can be written in the end of the table (Table-9)

Overall the paper is well written and conceptualized. Authors have given each and details regarding the research carried out.

6. PLOS authors have the option to publish the peer review history of their article (what does this mean?). If published, this will include your full peer review and any attached files.

Reviewer #1: No

Reviewer #2: No

Reviewer #3: No

---

## [Author Response · Author response to Decision Letter 0]

23 Feb 2021

Date: Feb 21 2021

Manuscript ID: PONE-D-20-39090

Menstrual Hygiene Practices among High School Girls in Urban Areas in Northeastern Ethiopia: A Neglected Issue in Water, Sanitation, and Hygiene Research

Corresponding authors: Metadel Adane (PhD)

Dear Dr. Srinivas Goli, (Ph.D)

Academic Editor

PLOS ONE

Thank you for your letter dated 19 Jan 2021 with a decision of revision required. We were pleased to know that our manuscript was considered potentially acceptable for publication in PLoS ONE, subject to adequate revision as requested by the reviewers, academic editor and the journal. Based on the instructions provided in your letter, we uploaded the file of the rebuttal letter; the marked up copy of the revised manuscript highlighting the changes made in the original submitted version and the clean copy of the revised manuscript. 

We have revised the manuscript by modifying the abstract, introduction, methods, results, discussion and other sections, based on the comments made by the reviewers and using the journal guidelines. Accordingly, we have marked in red color all the changes made during the revision process. Appended to this letter is our point-by-point response (rebuttal letter) to the comments made by the reviewers. 

We agree with almost all the comments/questions raised by the reviewers and provided justification for disagreeing with some of them. We would like to take this opportunity to express our thanks to the reviewers for their valuable comments and to thank you for allowing us to resubmit a revision of the manuscript. 

I hope that the revised manuscript is accepted for publication in PLoS ONE. 

Sincerely yours,

Metadel Adane (PhD) 

Response to the Journal Requirements Questions 

Question #1: Please ensure that your manuscript meets PLOS ONE's style requirements, including those for file naming.

Response: Thank you for this remark. We re-formatted the revised manuscript using the PLoS ONE format guidelines. The whole content of the manuscript, including the abstract, introduction, methods, discussion and reference are formatted using the guidelines (please see the revised version for each section).

Question #2: Please include additional information regarding the survey or questionnaire used in the study and ensure that you have provided sufficient details that others could replicate the analyses. For instance, if you developed a questionnaire as part of this study and it is not under a copyright more restrictive than CC-BY, please include a copy, in both the original language and English, as Supporting Information.

Response: We provided the questionnaire in English version and original language (Amharic) and as supporting information S II and S II, respectively. 

Question #3: You indicated that you had ethical approval for your study. In your Methods section, please ensure you have also stated whether you obtained consent from parents or guardians of the minors (<18 years old) included in the study or whether the research ethics committee or IRB specifically waived the need for their consent.

Response: Written consent was obtained from the participants age 18 and above and for study participants whose age were less than 18 years, parental assent consent was obtained. This also clearly added to revised version of the manuscript. 

Response to Reviewer Comments

Reviewer #1

The authors attempted to explore menstrual hygiene practices among high school girls in an urban setting of North-eastern Ethiopia. The study has several issues with the research design. The research design is not appropriate and, therefore, needs significant revision. How a study conducted in the urban area can have sample from rural areas. The authors have chosen five schools; however, the sample size from school varies drastically. What was the basis for choosing respondents from various schools? The formation of certain variables is not up to the mark. The close-ended questions that have limited options have the potential to include other options to choose from—thus limiting the study potential. The study failed to write a proper conclusion section. The study is full of limitations yet failed to include limitations in the text. The sampling procedure is confusing. The authors failed to write about sampling correctly. The study failed to impress, as it has a poor research design. Maybe the research design is not so poor- possibilities that authors failed to describe it adequately—suggested that the author describe the research design adequately. It is not clear whether the consent was written or verbal as information related to both modes of consent is given in text separately. Further, how distributing sanitary pads affected the response can be discussed. The article is written in standard English, however, failed to impress with its study design.

Response: Dear reviewer, we really appreciate your comments on the content of our paper. The comments are duly acknowledged and accepted. 

The study was conducted in the urban settings, where all schools situated in the city. However, for your concern of how the study was conducted, all peri-urban areas have no schools of their own. A small fraction of the study participant in our study lived in peri-urban areas attending classes in the urban schools. They travel up to 10km each day. So we did the revision in Table 1. Both urban and peri-urban areas are under Dessie City administration. 

Twenty-two study participants (4.1%) lived in the peri-urban and 95.9% areas of Dessie. Thank you for your key comment. In the footnote of Table 1, we noted this issue for clarification. 

The five schools were randomly selected among the 13 schools to meet the probability sampling criteria. The number off enrolled students in the 5 selected schools varied from 835 to 53 which leads to enormous variation in the sample size allocation (We provided the details of sample size allocation among schools and their grades in Fig 1 and supporting information SI I files under Sampling technique/procedure (Please see the Methods section on page 8 and Fig 1 and S I.). Due to the tuition fee in private schools, most students attend classes in public schools. This also leads to enormous variation in the sample size allocation. 

The study variables were selected after reviewing the literature. As we noted in the Methods section under the Data collection tool and procedure, the data were collected using a structured, interviewer-administered pre-tested questionnaire adapted from different relevant publications and tools prepared by UNICEF for assessing MHPs (Belayneh and Mekuriaw, 2019, Tegegne and Sisay, 2014, Upashe et al., 2015). The questionnaire was adapted from other studies conducted in Ethiopia (Belayneh and Mekuriaw, 2019, Tegegne and Sisay, 2014, Upashe et al., 2015). 

Being exclusively a quantitative study was one of our limitations; the close-ended questions have limited options. Thus we added possible options in the questions by reviewing different literatures (Please see the questionnaire as supporting information SI II and SI III). The conclusion part has been modified and incorporated in the revised version of the manuscript (See the conclusion in the abstract and below the discussion) and the limitations of the study also revised accordingly. We simplify the sampling procedure and hope it will clear the confusion. We also adequately describe the research design on the revised version.

Written consent was obtained from all participants above the age of 18 years. For participants under 18 years of age, assent was obtained from their parents or guardians (See in page 14 from lines 285 to 287). Since the distribution of the sanitary pads was done after the interviews, it had no effect on their responses. 

Overall, your comments are well taken. Thank you. 

Reviewer # 2

Abstract:

Question #1: - Abstract seems clumsy. Background can be improved. The authors failed to build a proper rationale in the background section. The third line of the abstract shall not start with “but”. ‘Consider rewriting it.

Response: Thank you for your comment, we modified the background section based on your comment. Please see in page 2 and 3 and also from lines 23 to 25 for the background improvemnt. 

Question #2: - The heading in the abstract – Methods, Results, and Conclusion must be assigned to a different line. Currently, the heading is placed at the end of the sentence.

Response: Yes, the heading for each section has its own line number (please see the revised version of the abstract and heading and sub-heading of the manuscript. 

Question #3: - Why menstrual hygiene practice is higher among girls whose mothers had secondary level education than those whose mothers had only primary level education; when menstrual hygiene practice is lower among girls whose mothers had college-level education than those whose mothers had secondary level education. Need to elaborate on this finding.

Response: This might be due to the fact that those college level educated mothers had less time to follow up with their daughters due to lack of time, whereas secondary level educated mothers ware mostly housewives who tended to advice their daughters. As indicated in our study, having literate mothers had a positive implication on girl’s menstrual hygiene practice. The discrepancies among literacy levels may emanate from the proportion of time mothers spent in the house versus time spent on to work. For example, mothers who had college level education spent more time in governmental office than at home. This effects the contact time of daughter-mother. However, as you can see in Table 9, our findings showed that both secondary level and college level education of mothers had positive implications for good MHP compared to illiterate mothers. 

Question #4: - The conclusion is erratic. Implications are not concrete. Moreover, keeping point number 3 cited above in mind- what level of education among mothers may be significant, as discussed in the study conclusion?

Response: Thank you for your comment. The conclusion part has been modified and incorporated in the revised version of the manuscript. As indicated in our study, overall, literate mothers had a positive influence on girl’s menstrual hygiene practices. The discrepancies among the literacy level may emanate from spending much time in the house over spending much time on to work. For example mothers who had college level education mostly spent their time in governmental office than their home. This consequently have an effect on the contact time of daughter-mother. 

Declaration Section:

Question #5: - In the ethics statement, the authors stated that 90 disposable sanitary pads were distributed to those individuals who were menstruating and could not afford them. Did the authors ask the respondents about their ongoing menstruation? If yes, how authors concluded that they could not afford the sanitary pads? Further, what implications this may have on the study design as it is clear from the beginning that such respondents were poor (Since they cannot afford sanitary pads)? Moreover, the distribution of sanitary pads not influencing the respondents‘ decision to participate in the study.

Response: We appreciate your comment. The distribution of the sanitary pads was done to meet ethical criteria. After the completion of each interview, the interviewer asked whether a study participant was on menses or not? Then yes and no access for sanitary pad, the pad will be dispensed. Since the dispensing done after the interview, it has nothing to do with their participation. Due to funding limitation we could give pads to only 90 girls.

Question #6: - Authors stated that all relevant data are within the manuscript and its supporting files. However, I could not find any relevant data associated with this submission. 

Response: Sorry for not attaching the data set during the original submission. We attached the data as supporting information (See S IV). 

Manuscript: 

Question #7: - Line number 94-95; the reference in the study context is a bit old. Try citing the latest reference and current prevalence.

Response: the reference has been updated. 

Question #8: - Authors have used good menstrual hygiene practices ‘and ‗menstrual hygiene practices‘ interchangeably, which is confusing.

Response: those interchangeably used words have been corrected and please see the revised version of the manuscript. 

Question #9: - Introduction is nicely articulated; however, it needs a bit of editing for ease of flow.

Response: we modified the introduction part (See the revised version of the introduction). 

Question #10: - Authors have mentioned in the abstract as well as in the manuscript text that the questionnaire was pre-tested. However, the information regarding the same is not given in full. When was it tested? Under whom guidance it was tested? The population upon which it was tested? - and other such details are missing. 

Response: all those missed data have been add to the edited version 

Question #11: - Did authors use 5 percent of margin of error conveniently or reached after consensus among various authors or some other relevant authority?

Response: We considered a prevalence of good MHP about 57% based on other studies and during the prevalence from 50% and around 50-60%; using of 5% margin of error is recommended for most epidemiological research. However, if the prevalence is about 40%, margin of error is 0.4, when around 30% margin of error is 0.3, when around 20%, margin of error 0.2 and when the prevalence around 10% margin of 0.1 is the most common approaches need for sample size determination. 

Question #12: - The study was conducted in an urban area of Dessie City. However, to calculate the sample size, the prevalence used was that of Adama town. It may be highly likely that menstrual hygiene practices' prevalence may be contrasting in these two cities.

Response: Thank you for your concern. During the previous study, where the prevalence taken, Adama were considered as a town. But currently Dessie and Adama are registered as cities in Ethiopia, with more than 100,000 population each. They also share common sociocultural settings (i.e. religion, community life style etc.). 

Question #13: - Representation of ‗z‘ in the sample size formula needs to be written in full for a general understanding of the readers. 

Response: Thank you for this key comment. It is re-written and see in page 7 form lines 140. 

Question #14: - How do authors reach the distribution of sample size between public (467) and private (79) schools? It is stated that distribution was proportionally done? Please elaborate on the proportion used.

Response: The Five schools randomly selected out of 13 schools to meet probability sampling requirements. Of the five schools, 2 of them were private and 3 schools were public. Most students attend classes in tuition-free public schools. So the actual student numbers in the public overweighs the private one. The proportional allocation of samples to each grade and their respective sections has been presented in table; please find in supporting files and figures (SI 1 and Fig 1). 

Question #15: - The sampling procedure is confusing. Consider presenting it through a flow chart. The authors stated that it was a two-stage sampling procedure; however, it is confusing in detailed text.

Response: Sorry for the confusion, the sampling procedure modified and presented and attached the flow in Fig. 1 and SI I. 

Question #16: - There is a contradiction in the statement by the authors. Somewhere, written consent was used, and at some places, verbal consent (Line- 199) was used in the text. Was the consent written or verbal or a mix of both- clarify. If verbal, how does the authenticity of the same can be produced? 

Response: Written consent was obtained from all participants whose age 18 years and above. For participants under 18 years of age, assent obtained from their parents or guardians, in addition to the guardians/parents’ assent was obtained from the participant themselves (See the revised version of the ethical issues in page 14 from lines 284 to 287). 

Question #17: - It is suggested to give the number of field investigators used for data collection. Were the same sets of investigators used for all the schools, or different schools were covered by different investigators? Data collection was carried out simultaneously in various schools, or was it collected one after the other? 

Response: Data collection was carried out by 3 supervisors, 5 female midwifery professional data collectors, and the principal investigator. We used the same supervisors for all the schools and the data collection was carried out the same day by assigning one data collector for each school for preventing information flow among students. (See the revised version in page 11 from lines 226 to 228). 

Question #18: - Direct observations were made to assess the suitability of the WASH facilities (Line- 201-202). How do investigators conclude the suitability of WASH facilities?

Response: An observational checklist by UNICEF and EMORY University clearly stipulated “when to say a school is convenient to manage menses”

Girl-friendly WASH facilities in school includes:

• Gender-specific, well kept, safe, clean and accessible sanitation facilities.

• Availability of uninterrupted water supply for 5-7 days.

• The continual availability of consumables, with particular attention on soap, water and culturally appropriate MHM materials.

• Disposal waste bins inside the latrines for discarded pads or their sanitary materials.

Please the revised version in the operational definitions of page 9 from 175 to 187-

Question #19: - Line number 206-207. Unpublished and published research was reviewed. Kindly elaborate on what kind of unpublished literature? 

Response: Unpublished literature included mostly government reports about MHP by the Ethiopian Ministry of Health and NGOs working on MHP. However, since our tool was developed based on published evidence, we deleted unpublished. Thank for this comment. 

Question #20: - It is suggested to give the equation for the statistics used in the study.

Response: We used binary logistic regression model and providing the formula of the model is not as such relevant for readers. In public health research, providing model education is not common, however, it is common for research by statisticians and math’s experts. Nevertheless, we can provide the formula by the next round of revision, if necessary. 

Question #21: - The study design is having severe issues. The number of respondents varies from 13 students in a school to 204 students from other school. Moreover, students were chosen from only two classes- 9th and 10th standard- Then, under what circumstances, it is possible that the age of the respondents varies from 13-19 years. It will be better if age-wise classification is presented in table 1. 

Response: We appreciate your concern, the number of respondents’ variation was explained in the above (#14) section. Due to tuition fee in private schools, most students attend classes in the public ones. So the actual student numbers in the public overweighs the private one. This leads to enormous variation in the sample size allocation. The two extreme ages (13 & 19) only represent only 2.9 % from all respondents in this age group. Most of the students category (80.8%) were 15-16 years old. The small proportion of extreme values was the result of;

A. starting school at early age

B. starting school at late age 

C. students who failed the grade 10 national examination had to repeat classes.

Question #22: - Study title clearly state that the study is conducted in urban area. Also, the methodology states the same. Then how is it possible to have a rural sample in the study? Are these respondents traveling from rural areas to urban schools? 

Response: Thank you again for this comment. Of course they are travelling from peri-urban areas to urban schools. They travel up to 10 KM from their homes in peri-urban areas to schools in the city each day. We explained this in the footnote of Table 1. We changed rural to peri-urban since Dessie city administration consists of peri-urban areas and urban areas (See Table 1). 

Question #23: - Authors stated that the response rate was around 98 percent. The study says that 98 percent of the respondents responded to all the questions. If so, how can 25 percent of the respondents have not recorded their answer (no response) for the marital status category? Is it like 98 percent agreed to record their responses and then left unanswered questions leading to an incomplete questionnaire?

Response: 25% of the students were afraid and shy to state their marital status due to complicated, broken relationships or unwanted marriages. So in order not to make the interview process tense, we added the option afraid to state. The amendment of the option was done after we performed the pre-test and by reviewed the literature. Thus we accepted the response being ‘afraid to state or mention” as an option for marital status. 

Question #24: - Table 3- option related to What is menstruation and other such questions may also have various other options than provided in the questionnaire. The questions asked might have included other categories also. 

Response: One of our study limitation was being a quantitative study only. Consequently close–ended questions forced them to select the options available. But to cope this we explored different previous both qualitative and quantitative studies and we tried to include all possible options for each questions.

Question #25: - Table 4- why sample is different for each variable?

Response: Some questions depend on the previous question responses (i.e. if a sample responds “no” for the question of “whether she heard about menses before menarche?” the sample will not be able to answer the “what was your source of awareness”. all other sample variations happen with the same phenomena.

Question #26: - Table 4- Source of awareness about menarche- the total adds to 500, and it was stated that n is 500. I wonder that the sources cannot be single for such information. Girls may receive information about menarche from both parents as well as their friends simultaneously and also from the media. Single source of information is undermining the study potential.

Response: Actually they may hear information from different source but for easy of recommendation, it’s better if we get the very first and most frequent source, despite of the likelihood of multiple sources

Question #27: - Table 4- With whom do you communicate frequently?- The question is confusing- communication for what? 

Response: The data collection was interviewer- based, hence they asked the interviewees “with whom do you frequently communicate in your family?” Also the question comes after “do you communicate about menstruation with your family?”. Thus confusion may be minimal with the use of trained data collectors. 

Question #28: - Table 5- water source functionality in the school- It is confusing how water functionality can be fewer than two days or 2-4 days in a school. The water functionality might be regular, and only in case of some service disruption, the water functionality in the school may be an issue. 

Response: Water source functionality was assessed based on personal/sample experience which helped us to evaluate the effect of WASH facilities on to a student. Most of the students (74%) stated that water was available 5-7 day, (26%) of the students experience shortage of water. The possible reasons for this could be, 

• During their search for water, other than at break time, they may get nothing. Our observational findings strengthen the idea. 

• During break-time, they may reach the water points when the reservoir was empty. 

Question #29: - Table 5- When student is allowed to use latrine- The question has limited options. It is not possible that every time a respondent may be allowed to go to the latrine either in the break. It may also depend upon the ongoing situation/lecture importance in school. Sometimes, teachers may or may not allow students to go to latrine. So, the options are not correct. 

Response: Irrespective of ongoing learning/lectures, a girl should be allowed to visit a toilet especially during her menses. Unless otherwise she may be unable to attend her class properly, her cloths may stain with blood or she may need to change her pads. Forcing a girl to go to toilet at specific times may have thus effect proper menstrual hygiene practice. 

Question #30: - Result found that 517 (96.5%) of the respondent use absorbent material. Moreover, all of these respondents change their absorbent at least once daily. At other places, the authors noted that they distributed certain sanitary pads as respondents could not afford

one. How can this be the case when 96 percent of the respondent are using and changing their absorbents daily? 

Response: As we previously responded, the distribution of the sanitary pads was done for the study participants who had no sanitary pads for alternative use or were unable to afford them during menstruation. Furthermore, 44 study participants were either using homemade absorbents or not using any at all, which indicates a need for sanitary pads, which we provided. Besides the remaining 46 pads were given to participants who complained about the cost of pads. We explained this on the ethical issues page 14 from lines 290 to 293. 

Question #31: - In the odds ratio table 9- the age-group can be divided into two groups only; 13-15 and remaining.

Response: the age classification was done after reviewing similar literature and for ease of discussion and recommendation purpose only. Under the current age category, we can discuss, compare and contrast with similar studies as you can see in the discussion section. Since in the multivariable logistic regression model, the age category was significantly associated good MHP, we kept that category within the paper. If we change the category as you suggested, the data analysis would have to be done again. 

Question #32: - What is not applicable category in marital status?

Response: 25% of the students were afraid (shy) to state their marital status due to complicated, broken relationships or unwanted marriages. So in order not to make the interview process tense, we added the option being “afraid” after we performed pre-testing and reviewing previous literature. Thus we considered being afraid as an a not applicable option for marital status. Make sure this is correctly worded

 Question #33: - Why father‘s occupation is not significant in the odds ratio in table 9. It is understood that if a father is working, it is more likely that a girl may afford sanitary napkin. Elaborate on this finding in the context of this study. 

Response: Thank you for your concern. Theoretically, a father who has income would buy sanitary pads. But in reality, this depends in Ethiopia on:

• the father’s knowledge of to menses (i.e. at least menarche age)

• the cost of sanitary pads

• Periodically allocated money for groceries may not include sanitary pads and also depends on the mother’s use of money.

Question #34: - Water source functionality in the school is not significant in the odds ratio model- elaborate on this finding. 

Response: Water source functionality may have an effect on to MHP but in our study it was not significant. This may be because girls may cope with water the shortage (i.e. they may used plastic bottles filled with water at periods of water unavailability). Furthermore, the number of water sources for each school is the same, which did not show variation in the data within the school 

Question #35: - The authors shall include the strength and limitation of this study.

Response: We thank for this pertinent comment. The strength and limitations have been included in the edited version of the manuscript (See the revised version of the manuscript in page 25- from lines 523 to 531). 

Question #36: - Conclusions are erratic. The study did not include the disabled; however, in the conclusion section, the authors propose policies for the disabled.

Response: since we did not use purposive sampling (we used probability sampling) the disabled were missed by chance. But when we reviewed the UNICEF/EMORY university girl’s friendly WASH facilities checklist, WASH services in all schools should have access for the disabled. Our observational studies found that the WASH facilities were not considering disabled students. To avoid confusion for readers, we avoid the recommendation about the disabled although our observational data showed that no school has facilities for the disabled. Please see the revised version of the conclusion and the recommendation, which is updated as per your point of view. 

Question #37: - It Seems that conclusion is not relevant to the study objective and just written haphazardly after going through certain available literature.

Response: We updated the conclusion carefully based on our findings from the multivariable analysis and observation data. Please it again in page 26 from lines533 to 547.

Question #38: - Please define what is environmental friendly sanitary napkin as discussed in conclusion section and why authors are proposing that special attention is to be given to lower grade levels students when the study population include 9th and 10th grade students?

Response: We agreed your concern and we moved the sentence to the limitation section and recommended further studies to identify environmentally friendly sanitary napkins. However, environmentally friendly sanitary napkins includes

• reusable sanitary cloth pads or

• being biodegradable

Regarding the lower grade, we mean grade 9th. But to minimize confusion, we deleted from the recommendation as you can see in the revised version. Our result showed that higher grade level (10th grade) has better menstrual hygiene practices than lower grade levels means (grade 9), so special attention should be given to the lower grade (grade 9). 

Question #39: - On the other hand, the use of drugs for irregular menses, especially contraceptive pills, may currently not be feasible in Ethiopia due to negative cultural attitudes of parents toward their use- How does this sentence used in conclusion section is relevant to the study context?

Response: we revised the conclusion and your concern is no longer within the paper. Your comment is well taken. 

Reviewer # 3: 

Introduction

Question #1: - Overall the introduction is well-written and covered all domains that need to be highlighted in the Introduction. I request authors to please write hypothesis in the last para of the Introduction.

Response: Dear reviewer, we would like to thank you for the valuable comments and we research questions at the introduction. We did not used hypothesis during the research rather we used research questions which are included at the end of the introduction on page 6 form lines 110 to 115. 

Results

Question #2: - Please do not highlight the text in page -18 line 357

Response: we edited based on your comment. Thank you. 

Discussion

Question #3: All the findings are well discussed in the respective section. Just a small suggestion where ever ―West Bengal‖ is being written, please write in brackets (India) as far as I know it‘s the same region which is from India.

Response: we revised based on the comment. 

Question #4: - I request the authors to write one para (3-4 lines) as limitations and strength of the study. Please write that in the end of the discussion

Response: We add our limitations as per the request, but the nature of cross-sectional studies, it would have been difficult to explain about the strength of the study (See the limitations in the revised version in page 25). 

References

Question #5: - Some references do not have publisher‘s name. Please do amend those references.

Response: Thank you we updated those that have information on publishers. 

Tables 

Question #5: - The highlighted estimates are not needed. I mean do not highlight anything in the tables. COR and AOR full forms can be written in the end of the table (Table-9)

Response: Thank you, we updated as suggested without highlighting. 

I hope that the revised manuscript is accepted for publication in PLoS ONE. 

Sincerely yours,

Metadel Adane (PhD in Water and Public Health)

---

## [Decision Letter · Decision Letter 1]

4 May 2021

Menstrual Hygiene Practices among High School Girls in Urban Areas in Northeastern Ethiopia: A Neglected Issue in Water, Sanitation, and Hygiene Research

PONE-D-20-39090R1

Dear Dr. Adane,

We’re pleased to inform you that your manuscript has been judged scientifically suitable for publication and will be formally accepted for publication once it meets all outstanding technical requirements.

Kind regards,

Srinivas Goli, Ph.D.

Support Staff - Editorial

PLOS ONE

Additional Editor Comments (optional):

Considering favourable opinions from the reviewers, I am going with a decision of Accept.

Reviewers' comments:

Reviewer's Responses to Questions

**Comments to the Author**

1. If the authors have adequately addressed your comments raised in a previous round of review and you feel that this manuscript is now acceptable for publication, you may indicate that here to bypass the “Comments to the Author” section, enter your conflict of interest statement in the “Confidential to Editor” section, and submit your "Accept" recommendation.

Reviewer #1: All comments have been addressed

Reviewer #3: All comments have been addressed

2. Is the manuscript technically sound, and do the data support the conclusions?

Reviewer #1: Yes

Reviewer #3: Yes

3. Has the statistical analysis been performed appropriately and rigorously? 

Reviewer #1: Yes

Reviewer #3: Yes

4. Have the authors made all data underlying the findings in their manuscript fully available?

Reviewer #1: Yes

Reviewer #3: Yes

5. Is the manuscript presented in an intelligible fashion and written in standard English?

Reviewer #1: Yes

Reviewer #3: Yes

6. Review Comments to the Author

Reviewer #1: The authors have carried out all the revisions carefully and the same has been visible from the revised version of manuscript.

Reviewer #3: (No Response)

7. PLOS authors have the option to publish the peer review history of their article (what does this mean?). If published, this will include your full peer review and any attached files.

Reviewer #1: **Yes: **Ratna Patel

Reviewer #3: No

---

## [Editor Report · Acceptance letter]

27 May 2021

PONE-D-20-39090R1 

Menstrual Hygiene Practices among High School Girls in Urban Areas in Northeastern Ethiopia: A Neglected Issue in Water, Sanitation, and Hygiene Research 

Dear Dr. Adane:

I'm pleased to inform you that your manuscript has been deemed suitable for publication in PLOS ONE. Congratulations! Your manuscript is now with our production department. 

Kind regards, 

on behalf of

Dr. Srinivas Goli 

Academic Editor

PLOS ONE